# Behind the Hospital Ward: In-Hospital Mortality of Type 2 Diabetes Mellitus Patients in Indonesia (Analysis of National Health Insurance Claim Sample Data)

**DOI:** 10.3390/ijerph21050581

**Published:** 2024-05-01

**Authors:** Ede Surya Darmawan, Vetty Yulianty Permanasari, Latin Vania Nisrina, Dian Kusuma, Syarif Rahman Hasibuan, Nisrina Widyasanti

**Affiliations:** 1Faculty of Public Health, Universitas Indonesia, Depok 16424, Indonesia; vetty.yulianty@ui.ac.id (V.Y.P.); latin.vania@ui.ac.id (L.V.N.); syarif.rahman@ui.ac.id (S.R.H.); nisrina.widyasanti@ui.ac.id (N.W.); 2Center for Health Policy and Administration Studies, Faculty of Public Health, Universitas Indonesia, Jawa Barat 16424, Indonesia; 3Department of Health Services Research and Management, School of Health & Psychological Sciences, City University of London, London EC1V 0HB, UK; dian.kusuma@city.ac.uk

**Keywords:** diabetes mellitus, in-hospital mortality, risk factors

## Abstract

The rising global prevalence of diabetes mellitus, a chronic metabolic disorder, poses significant challenges to healthcare systems worldwide. This study examined in-hospital mortality among patients diagnosed with non-insulin-dependent diabetes mellitus (NIDDM) of ICD-10, or Type 2 Diabetes Mellitus (T2DM), in Indonesia, utilizing hospital claims data spanning from 2017 to 2022 obtained from the Indonesia Health Social Security Agency or Badan Penyelenggara Jaminan Sosial (BPJS) Kesehatan. The analysis, which included 610,809 hospitalized T2DM patients, revealed an in-hospital mortality rate of 6.6%. Factors contributing to an elevated risk of mortality included advanced age, the presence of comorbidities, and severe complications. Additionally, patients receiving health subsidies and those treated in government hospitals were found to have higher mortality risks. Geographic disparities were observed, highlighting variations in healthcare outcomes across different regions. Notably, the complication of ketoacidosis emerged as the most significant risk factor for in-hospital mortality, with an odds ratio (OR) of 10.86, underscoring the critical need for prompt intervention and thorough management of complications to improve patient outcomes.

## 1. Introduction

The global incidence and prevalence of diabetes mellitus have demonstrated a consistent upward trajectory in recent decades [1]. Diabetes mellitus, characterized by elevated blood glucose levels, is a chronic metabolic disorder with various etiologies. Type 1 diabetes results from the autoimmune destruction of pancreatic beta cells, leading to insulin deficiency [2]. Conversely, type 2 diabetes typically arises from insulin resistance coupled with progressive beta cell dysfunction. Additionally, gestational diabetes, a temporary condition occurring during pregnancy, contributes to the spectrum of diabetes [2]. Insulin plays a pivotal role in facilitating the cellular uptake of glucose from the bloodstream for energy metabolism [3]. Dysfunction in insulin action results in persistent hyperglycemia, heightening the risk of long-term complications across all diabetes subtypes, encompassing cardiovascular disease, nephropathy, neuropathy, and retinopathy.

According to the International Diabetes Federation, the total number of diabetes cases worldwide in 2021 reached 537 million, equivalent to 10.5% of the adult population [4]. This figure has increased from 2014, when the World Health Organization (WHO) recorded 422 million people worldwide suffering from diabetes. It is estimated that this number will rise to 643 million by 2030 [4]. Type 2 diabetes is the most common form compared with other types (type 1 diabetes and gestational diabetes), accounting for over 90% of all diabetes cases [5]. The majority of diabetes sufferers come from low- and middle-income countries, with nearly half of them unaware that they are living with the condition. Many individuals are diagnosed with diabetes incidentally because the disease has progressed to a severe stage or is diagnosed too late. This is due to type 2 diabetes often emerging silently, even without symptoms, hence often being referred to as the ‘silent killer’ [6].

Indonesia falls within the category of middle-income countries, with diabetes ranking as one of the leading chronic diseases contributing to the highest mortality rates. According to the Institute for Health Metrics and Evaluation in 2019, diabetes ranked third among causes of death in Indonesia, with 57.42 deaths per 100,000 people [7]. Data from the International Diabetes Federation indicate a rapid increase in the number of diabetes patients in Indonesia over the past decade, with 19.5 million individuals or approximately 7% of the population suffering from diabetes in 2021 [4]. This places Indonesia fifth globally in terms of the highest number of diabetes cases. Additionally, there were 14.3 million undiagnosed cases of diabetes in 2021. In the same year, there were a total of 236,711 deaths attributed to diabetes in Indonesia. The mortality rate due to diabetes in Indonesia is projected to increase by 2030 [4]. The triggering factors for this rise include delays in diagnosing diabetes in patients, leading to the disease progressing to severe stages and resulting in complications. Complications of type 2 diabetes mellitus may arise if not managed properly. In the short or long term, this disease can adversely affect the function of almost every human organ [8]. Complications that may emerge include cardiovascular diseases, peripheral vascular diseases, diabetic retinopathy that threatens vision, diabetic neuropathy, and kidney complications [9]. A retrospective cohort study conducted in Hong Kong from 2010 to 2019 showed a trend of increasing kidney complications and mortality [9]. Poor outcomes in type 2 diabetes patients are mostly found in younger individuals, thus necessitating good control and quality healthcare services for younger type 2 diabetes mellitus patients. 

In Indonesia, the National Health Insurance or Jaminan Kesehatan Nasional (JKN) Program provides comprehensive health protection through a mechanism of social insurance based on mutual cooperation and affordable premiums [10]. The National Health Insurance Program enables the Indonesian population to access specialty care without financial difficulties, as all expenses are covered by the program. The program, initiated in 2014, is mandatory for all Indonesian citizens, including foreigners who have worked in Indonesia for at least 6 months and have paid premiums. This program is administered by the Health Social Security Agency (BPJS Kesehatan), which implements a tiered referral system to access healthcare services [11]. 

Healthcare services start from primary healthcare facilities known as FKTP acting as ‘gatekeepers’ and having the authority to refer patients to advanced referral healthcare facilities, namely hospitals known as FKRTL. Patients enrolled in the National Health Insurance Program with type 2 diabetes who have experienced complications require specialized care that is only available at referral healthcare facilities, namely hospitals [10]. These healthcare services are expected to control critical conditions of type 2 diabetes patients, allowing them to return home in stable condition and preventing severity and mortality. 

Accordingly, this study was undertaken to investigate the incidence of in-hospital mortality among hospitalized patients with type 2 diabetes mellitus while also exploring the determinants contributing to this outcome. The research endeavors to address a crucial gap in understanding the factors associated with mortality within this patient population, thus contributing valuable insights to the field of diabetes management. By elucidating the determinants of in-hospital mortality, this study aims to provide actionable information for healthcare practitioners to optimize patient care protocols and improve clinical outcomes. The findings of this research are anticipated to offer significant implications for enhancing the quality of care delivery and ultimately improving patient survival rates among individuals with type 2 diabetes mellitus.

## 2. Materials and Methods

This study utilized claim data from 2017 to 2022 provided by the Social Security Administration for Health or Badan Penyelenggaran Jaminan Sosial (BPJS) Kesehatan, which are openly accessible.

### 2.1. Introduction of BPJS Kesehatan Sample Claim Data

The Social Security Health Insurance Program or Program Jaminan Kesehatan Nasional (JKN) in Indonesia is a mandatory program for all Indonesian citizens, aimed at guaranteeing program participants funding to access healthcare services. Since 2018, the administrator of the JKN program, BPJS Kesehatan, has released sample data that are openly accessible for use by academics and researchers [12].

#### 2.1.1. Sample Selection Steps

The sample selection in the recent [2022] Sample Data originates from two sub-populations of BPJS Kesehatan participants, and an illustration of the sampling steps can be seen in the Figure 1:
Sub-population of participants who joined BPJS Kesehatan before 2022.Sub-population of participants who joined BPJS Kesehatan in 2022, consisting of new individuals who actively registered as BPJS Kesehatan participants in 2022 based on the master data from membership files until the end of 2022.

The process of selecting BPJS Kesehatan participant samples from this sub-population is carried out independently, using the following steps for each sub-population:
Preparing the Sampling Frame:Gather all sampling units, which are participant families, into a sampling frame.The sampling frame is taken from the participant database as of 31 December 2022.Building StrataStrata are constructed based on the combination of two variables: primary healthcare and family category (three categories). Family categories: Category 1: Participants who have received health services at FKTP; Category 2: Participants who have received health services at FKRTL; Category 3: Families where no members have received health services. If each FKTP has members from all three categories, there will be a maximum of three strata.Selecting Family SamplesEach stratum is randomly selected for a minimum of (N, 1) families, meaning one family unit is selected from each stratum.At this stage, the sampling process is completed and followed by filtering the complete (master) data for data selection based on the selected samples.Obtaining Participant Sample DataFiltering master participant data (database) using the family code criteria as the selected family code in step 3.Obtaining Service Claim Sample Data (from primary health care and hospital)Obtaining service data based on membership samples by filtering the service database using participant code criteria, including participant codes selected in step 4.

#### 2.1.2. Weight

BPJS Kesehatan claims sample data provide a ‘weight’ variable. The weight needs to be used for analysis using BPJS Kesehatan sample data to ensure that the obtained results represent the population. Individual weighting of selected samples from BPJS Kesehatan sample data is conducted to correct imperfections in the sample resulting in bias or differences between the sample and the population. Imperfections occur owing to non-uniform sampling probabilities resulting from the sampling design used. Sample allocation in each stratum is not given proportionally, so the chances for each family to be selected as a sample are not equally large.

Before calculating individual weights, family weights need to be calculated first. Mathematically, if *p_i_* represents the probability of the *i*-th family being selected, then the weight given to the selected family (*w_i_*) is:$$w_{i}=\frac{1}{p_{i}}$$

As an illustration, Table 1 presents an overview of the weight calculation for each family in six strata. For example, in stratum number 3 (which consists of FKTP 1 and Category 3), there is a population of 6200 families, and one family is sampled. Each family has a probability of 1/6200 or approximately 0.0002 of being selected as a sample. The sample weight for the family is 6200. This means that one family in the sample represents 6200 families in the population. The same applies to other strata.

Up to this point, the obtained weights are household weights. The next step is to calculate individual weights to ensure that the distribution of individual sample characteristics reflects the characteristics of the population individuals. Individual weights are obtained by applying household weights to each participating family member. Subsequently, weight adjustments are made by multiplying them by a constant based on the sample’s representation of the population, considering variables such as gender, age group, and participant segmentation. 

### 2.2. Research Design and Participants

This study employed an observational cross-sectional design conducted from January 2017 to December 2022 on patients undergoing hospitalization. The sample in this study comprised all participants of the Indonesia national health insurance program with non-insulin-dependent diabetes mellitus (NIDDM) diagnosis codes in the ICD-10 E11 or type 2 diabetes mellitus, in Indonesia, and aged ≥30 years from 2017 to 2022. After weighting, a total of 610.809 patients met the criteria and became samples in the study.

#### 2.2.1. Dependent Variable 

Our study focuses on the incidence of in-hospital mortality among type 2 diabetes mellitus patients undergoing hospitalization.

#### 2.2.2. Independent Variables

This study examines eleven variables: [1] comorbidities, [2] age, [3] sex, [4] case severity level, [5] hospital ownership, [6] hospital level, [7] hospital region, [8] insurance class, [9] employment status, [10] hospital location (urban/rural), [11] length of hospital stay, and [12] case claim. Employment status in this study is categorized into dependent employment, participants who have regular monthly income; independent employment, participants with irregular monthly income; and participants receiving government subsidy contributions. Severity level in this study is categorized as severe and mild. The severity level is input data from the hospital claims unit according to the diagnosis made by doctors based on the severity level of the case. Regions in this study are grouped according to the classification of the National Development Planning Agency (BAPPENAS). Region categories consist of Sumatra, Java and Bali, Kalimantan, Sulawesi and Papua, Nusa Tenggara, and Maluku. The insurance class in this study is the inpatient room class obtained by patients based on the amount of premiums paid. Insurance class categories are grouped into class 1, the highest; class 2; and class 3, the lowest. Hospital ownership in this study is differentiated into government-owned, private, military/police, and state-owned enterprise hospitals. Hospital classification in this study is differentiated into Class A hospitals with the highest competency, serving as national referral centers; Class B hospitals, which are provincial referral centers; Class C and Class D hospitals with lower competency levels; and hospitals providing specialized services.

### 2.3. Statistical Analysis 

This research will look at in-hospital mortality according to the conceptual framework that can be seen in Figure 2. The hypothesis in this study is that there is a relationship between existing independent variables and the incidence of in-hospital mortality. The study will conduct multivariable regression analysis to observe the in-hospital mortality determinant factors. A statistical significance level of 5% or lower is considered significant.

## 3. Results

### 3.1. Descriptive Statistics

The primary outcomes, samples, and hospital characteristics of the study are presented in Table 2.

Hospital discharge status is this study’s dependent variable, and when used to indicate mortality status, it shows that most patients were discharged alive, namely 93.4%, while 6.6% died. This indicates that the majority of cases handled ended with positive results.

Patient Characteristics

In terms of patient characteristics, 61.8% are female and 38.2% are male, with a wide age range from 30 to 96 years and a median age of 57, indicating most are adults to elderly. Employment status shows that 40.3% are independently employed, 37.1% are from subsidized groups, and 22.7% are dependently employed, reflecting diverse economic backgrounds. Insurance class distribution shows 52.8% in Class 3, 28.0% in Class 1, and 19.2% in Class 2, suggesting broader health service access among the lower classes.

A total of 10.3% of patients had comorbid disorders of fluid, electrolyte, and acid–base balance (ICD10: E87), indicating the prevalence of these disorders in the patient population. Additionally, 4.2% of patients were reported to have anemia (ICD10: D64), while urinary tract infection (ICD10: N39) occurred in 5.2% of patients. Comorbid pneumonia (ICD10: J18) was found in 3.7% of patients. These percentages reflect the distribution of several critical clinical comorbidities that affect some patients and are essential to consider in health management and hospital care strategies.

The majority of patients, 75.6%, did not experience any complications at discharge, suggesting effective management of most cases. However, complications still occurred: 4.6% experienced a coma (ICD10: E11.0), requiring intensive treatment; 3.7% had ketoacidosis (ICD10: E11.1), a life-threatening condition needing immediate care; and 4.9% had multiple complications, indicating complex cases. Additionally, 7.3% had other specified complications (ICD10: E11.6), and 3.9% had unspecified complications (ICD10: E11.8), showing diverse manifestations needing varied approaches. Severity-wise, 59.8% of cases were mild, 25.9% moderate, and 14.3% severe, with hospitals treating mostly less-severe cases.

2.Hospital Characteristic

In terms of hospital characteristics, 50.3% of patients were treated in private hospitals and 40.5% in government hospitals. Level C hospitals (46.9%) handle the most patients, followed by level B (30.1%) and D (17.9%), and level A, which are central referral hospitals, handle the least (3.4%). Most of the patients in this study came from hospitals in Java and Bali (61.3%), indicating a more centralized concentration of health infrastructure in this region.

The length of stay and case claim (in USD) variables illustrate that patient length of stay ranged from 0 to 46 days with a median of only 4 days, indicating short-term care is more common. Case claims ranged from USD 147 to USD 1089, with a median value of USD 300.09, suggesting that treatment costs were relatively affordable for the majority of cases.

The table presents the distribution of patient characteristics and hospital characteristics among those who were discharged alive and those who died during hospitalization.

Patient Characteristics

Based on the sex variable, women have a slightly higher in-hospital mortality rate (6.8%) than men (6.3%). For employment status, the subsidized group showed the highest in-hospital mortality rate at 9%, while dependent employment had the lowest mortality rate at 3.8%. In the insurance class, Class 3 experienced a high in-hospital mortality rate (8.1%), while Class 1 and Class 2 both had a mortality rate of 5%. In the comorbidities category, patients with disorders of fluid, electrolyte, and acid–base balance who had these conditions showed a higher rate of in-hospital mortality (10.3%) compared with those who did not (6.2%). For complications, NIDDM with ketoacidosis (E111) had the highest in-hospital mortality rate (29.8%), while NIDDM with other specified complications (E116) showed the lowest in-hospital mortality rate (2.6%). Based on the case severity level, patients with severe cases have a much higher in-hospital mortality rate (20.6%) than the in-hospital mortality rate in mild cases, which is only 3.7%.

2.Hospital Characteristics

Based on the hospital ownership variable, state-owned enterprise hospitals have the lowest in-hospital mortality rate (0.8%), while government hospitals have the highest mortality rate (9.4%). For hospital level, specialist services have the highest mortality rate (14.9%), and Level D has the lowest mortality rate (3.6%). Regarding the hospital region, Papua, Nusa Tenggara, and Maluku have the lowest mortality rate (2.9%), while Kalimantan shows the highest mortality rate (10%). Based on hospital location, the county has a higher death rate (7.7%) compared with the city (5.3%).

### 3.2. In-Hospital Mortality Determinant

Table 3 and Table 4 present the results of a multivariable logistic regression analysis examining the factors associated with in-hospital mortality among patients with non-dependent insulin diabetes mellitus.

Patient Characteristics

In a study of patients with type 2 diabetes mellitus, multivariable logistic regression identified several key factors affecting in-hospital mortality. Age increased the risk of death by 1.03 times per year (odds ratio 1.03, *p*-value < 0.05). Employment status also influenced outcomes, with the subsidized group facing a 1.84 times higher risk compared with dependent employment (odds ratio 1.84, *p*-value < 0.05), while independent employment decreased risk (odds ratio 0.89). Gender and insurance class showed no significant effect.

Regarding complications of type 2 diabetes mellitus, NIDDM with ketoacidosis (ICD10: E11.1) has the strongest association with mortality, with an odds ratio of 10.86, indicating an almost 11 times higher risk of death compared with patients without complications (*p*-value < 0.05). Other complications, such as NIDDM with coma (ICD10: E11.0) and multiple complications (ICD10: E11.7), also showed a significantly increased risk of death, with odds ratios of 4.48 and 3.56, respectively. Meanwhile, Non-insulin-dependent diabetes mellitus with other specified complications (ICD10: E11.6) is unique because it shows a reduced risk of death (odd ratio 0.74).

For comorbidities, disorders of fluid, electrolyte, and acid–base balance (ICD10: E87) show an increased risk of death of 4% compared with patients without comorbidities (odds ratio 1.04, *p*-value < 0.05). Meanwhile, comorbidities such as anemia (ICD10: D64) and urinary tract infection (ICD10: N39) actually show a reduced risk of death with odds ratios of 0.53 and 0.51, respectively. Most notable is pneumonia (ICD10: N39), which shows the most significant reduction in risk of death, with an odds ratio of only 0.30, representing a 70% risk reduction compared with no comorbidities.

For the case severity level, patients with moderate cases have an almost two-fold increased risk of death (odds ratio 1.93, *p*-value < 0.05) compared with mild cases, which are used as a reference. Severe cases have a very significant increased risk, with an odds ratio of 10.49, indicating a more than tenfold increased risk of death compared with mild cases.

2.Hospital Characteristics

Multivariable logistic regression analysis of hospital characteristics shows that various factors have a significant relationship with mortality in type 2 diabetes mellitus patients. In the hospital ownership category, government hospitals show an increased risk of death of 1.27 times compared with private, which is the reference (odds ratio 1.27, *p*-value < 0.05). Furthermore, military/police hospitals have an even higher risk, with an increase of 1.48 (odds ratio 1.48). In contrast, state-owned enterprises (BUMN) show a reduced risk of death with an odds ratio of only 0.19, which indicates a reduction in the risk of death by 0.19 times.

At the hospital level, there is an increased risk of death at Level A and B hospitals compared with Level D, with odds ratios of 2.60 and 2.31, respectively. Specialist services show the highest increased risk of death of all levels, with an odds ratio of 4.39, while Level C has a more moderate increased risk of 1.53. In terms of hospital region, Sulawesi shows the highest increased risk of death compared with Java and Bali e, with an odds ratio of 2.36. Other regions, such as Kalimantan and Sumatra, also experience an increased risk of death. However, Papua, Nusa Tenggara, and Maluku show a significant reduction in the risk of death (odds ratio 0.42).

For hospital location, hospitals located in the county have a 49% increased risk of death compared with those in the city e. Meanwhile, length of stay has an inverse relationship with mortality, where a one-unit reduction in length of stay is associated with a 0.81 times reduction in the risk of death (odds ratio 0.81, *p*-value < 0.05). There is no significant relationship between case claim (in USD) and mortality, showing a *p*-value greater than 0.05.

## 4. Discussion

### 4.1. Patient Characteristics:

#### 4.1.1. Sex

Even though this study did not find any association with in-hospital mortality, previous studies mention higher mortality rates among females than males, which can be linked to gender-based variations in disease presentation and management, as well as biological differences. Women may face unique challenges like hormonal fluctuations and pregnancy-related complications, affecting disease progression and treatment outcomes. Socio-cultural factors and access to healthcare also contribute to these disparities. Studies show that women with diabetes are particularly vulnerable to dying from respiratory tuberculosis and accidents, mainly falls [13]. Another study indicates that the increased in-hospital mortality of diabetes patients, especially females, is due to higher rates of cerebrovascular disease and infections [14]. Another shows women with diabetes have a greater risk of death than men with diabetes, with comorbid cardiovascular disease having a greater impact on women with diabetes than men, especially if diagnosed at a later stage [15]. This suggests that differences in disease manifestation, progression, and response to treatment between males and females potentially contribute to differential mortality rates. Further research is necessary to comprehensively understand the underlying reasons for this phenomenon and to develop targeted interventions to mitigate the disparity in outcomes among male and female diabetes mellitus patients.

#### 4.1.2. Age

The association between older age and increased odds of mortality is consistent with the well-established concept of age-related health deterioration. A previous study indicates that diabetes mellitus is associated with increased mortality in older age due to its impact on the cardiovascular system [16], while another study shows that cerebrovascular accident and infection are major causes of mortality in older patients [17]. This shows that older patients are more susceptible to comorbidities or complications that can worsen the condition and increase in-hospital mortality.

#### 4.1.3. Employment Status

The higher mortality rates among diabetes mellitus patients in the subsidized group under Indonesia’s National Health Insurance (JKN) relate to several factors. These patients often have a lower socioeconomic status, linked to a higher prevalence of risk factors like unhealthy diets, sedentary lifestyles, and limited access to preventive healthcare. Socioeconomic disparities affect their ability to afford essential medications and healthy food, complicating diabetes management. Additionally, limited health literacy and awareness can delay medical attention and lead to inadequate self-care and poor treatment adherence, resulting in uncontrolled diabetes and higher mortality. These groups also face barriers to accessing quality healthcare due to financial constraints and infrastructure disparities, hindering access to specialized care and regular monitoring, which further increases the risk of complications. Previous studies also show a correlation between socioeconomic status and in-hospital mortality for diabetes patients [18,19].

Overall, addressing these disparities in healthcare access, socioeconomic status, and health literacy is crucial for improving outcomes and reducing mortality rates among diabetes mellitus patients, particularly those in subsidized groups covered by Indonesia’s National Health Insurance (JKN). This necessitates targeted interventions such as health education programs, community-based interventions, and healthcare system reforms aimed at enhancing equity and accessibility to comprehensive diabetes care for all segments of the population.

### 4.2. Complications, Comorbidities, and Case Severity

Diabetes mellitus alone, without any accompanying health issues, was not associated with increased mortality or reduced quality of life among the general population of Intensive Care Unit (ICU) patients. However, the presence of complications or comorbid conditions significantly reduced patient survival rates [20].

#### 4.2.1. Complications

In this study, we found that there were two specific complications associated with an increased risk of in-hospital mortality in type 2 diabetes mellitus patients, namely ketoacidosis and coma, with odds ratios (ORs) of 10.86 and 4.48, respectively. Scientifically, patients with ketoacidosis have a higher risk of in-hospital mortality because ketoacidosis creates an extreme metabolic imbalance, which can lead to severe dehydration, electrolyte balance disorders, and acidosis that requires intensive medical intervention for stabilization. These findings support previous evidence showing that the incidence of hospital admissions for diabetic ketoacidosis is growing over time, particularly in patients with type 2 diabetes mellitus. However, the duration of hospitalization has decreased [21]. Previous studies also stated that diabetic ketoacidosis is the main cause of death among children and young adults with diabetes [22]. Factors contributing to this risk include inadequate glycemic monitoring in diabetes clinics, deficiencies in diabetes education, and alcohol consumption. Inadequate glycemic monitoring often results in inappropriate management of fluctuating blood glucose levels, while deficiencies in diabetes education can hinder patients’ ability to manage their condition effectively. Furthermore, alcohol consumption is known to worsen metabolic control, increase the risk of hypoglycemia, and disrupt patients’ capacity to manage their insulin therapy.

Likewise, type 2 diabetes mellitus progressing to coma also presents a significant risk. Coma in the context of diabetes is often associated with extreme hyperglycemia or hypoglycemia, both of which can cause damage to the brain and other organs, increasing the risk of in-hospital mortality [23,24]. These two conditions, ketoacidosis, and coma, highlight the complexity and severity of type 2 diabetes mellitus as a metabolic disease with systemic implications that can be life-threatening, requiring a rapid and appropriate management approach.

#### 4.2.2. Comorbidities

The multivariable logistic regression analysis found that patients with disorders of fluid, electrolyte, and acid-base balance as secondary diagnosis had an odds ratio (OR) of 1.04 for experiencing in-hospital mortality. Although these OR values are relatively small, indicating only a modest increase in risk, it is important to consider the clinical and biological context in which these values operate. A previous study shows that patients with type 2 diabetes mellitus exhibit significant changes in serum metabolites, including amino acids, which may reflect complications and disease progression [25]. Other studies also show that acid–base and electrolyte disturbances are common in patients with type 2 diabetes mellitus, including the prevalence of alkalosis and metabolic acidosis, which can occur despite normal renal function [26]. Furthermore, fluid shifts, electrolyte imbalances, and acid–base disorders are factors causing complications such as diabetic ketoacidosis in patients with type 2 diabetes mellitus, which can cause multi-organ failure [27]. This indicates that these disorders are a common phenomenon in these patients and can develop in the absence of severe renal dysfunction. Therefore, although the odds ratio recorded in this study is small, there are significant clinical implications that support the idea that comorbid variables of fluid, electrolyte, and acid–base balance disorders have a strong association with the emergence of ketoacidosis in diabetic patients, which can lead to in-hospital mortality.

This study indicates that patients with a secondary diagnosis of anemia have a lower risk of in-hospital mortality, as evidenced by an odds ratio (OR) of 0.53 with a 95% confidence interval ranging from 0.5 to 0.56 and a *p*-value of less than 0.05. This finding appears to contradict previous research that links anemia with an increased risk of death in type 2 diabetes mellitus patients [28,29,30], suggesting a need to explore the reasons for this discrepancy. Further investigation employing a detailed analysis of the methodologies used and possibly conducting additional studies with different designs or in varied settings may be necessary to clarify these conflicting findings and better understand the impact of anemia on mortality in patients with type 2 diabetes mellitus.

Similarly, the finding from this study indicating an odds ratio (OR) of 0.51 for urinary tract infection (UTI) occurrence in patients with type 2 diabetes mellitus suggests that there may be a protective factor or an artifact of study design that is influencing the observed association, and previous studies suggest an elevated risk and prevalence among this group [31,32]. There are a few potential explanations and arguments to consider. Inadequate control of confounders, such as age, sex, medication use, or renal impairment, could misrepresent the association. Alternatively, the analysis method might lead to over-adjustment for variables like hyperglycemia, reducing the observed effect size. However, there could genuinely be a protective factor at play. For example, regular medical supervision and proactive management of health in diabetic patients could lead to early detection and treatment of UTI symptoms, reducing the apparent prevalence or severity of UTIs in this group [33]. Additionally, certain medications commonly used in T2DM, such as SGLT2 inhibitors, were studied for their potential impact on renal glucose handling, which could inadvertently affect the urinary environment in a way that reduces UTI risk. 

Based on the statistical calculation, type 2 diabetes mellitus patients with a secondary diagnosis of pneumonia had an odds ratio (OR) of 0,3 for experiencing in-hospital mortality. This aligns with a study that says diabetes is associated with lower in-hospital mortality after community-acquired pneumonia (CAP) hospitalization [34]. However, although patients with type 2 diabetes mellitus who experience CAP have slightly lower in-hospital mortality compared with patients without diabetes, this does not indicate that pneumonia lowers their risk of death. It rather suggests that, despite the higher risk of CAP incidence, the outcomes for these patients may have been improved, or diabetes per se might not increase the risk of mortality from CAP as much as previously anticipated. To sum up, the reduction in mortality may be more associated with improvements in hospitalization care and comorbid conditions in general over the study period, rather than pneumonia specifically providing a protective effect against patients with diabetes.

#### 4.2.3. Case Severity

In the BPJS Kesehatan dataset, the case severity variable reflects patient condition complexity. Patients classified as severe—with numerous significant complications or comorbidities—face substantially higher in-hospital mortality risks. The strikingly elevated odds of mortality among patients with severe cases highlight the critical importance of the early detection of and strategic intervention for high-risk individuals. The comprehension of case severity enables healthcare professionals to deliver personalized and efficient treatment strategies, thereby enhancing patient outcomes.

Determined by the presence and quantity of comorbidities entered by hospital claims or case-mix units as approved by specialists, this variable impacts claims calculations under the premise that greater severity necessitates increased treatment expenditures. Ultimately, incorporating the case severity variable into BPJS health data aids in precise claims calculation and is pivotal in clinical management. Early identification of high-risk patients during hospitalization allows healthcare facilities to allocate and optimize resource distribution and personalized interventions, mitigating mortality risks and improving treatment outcomes.

### 4.3. Hospital Characteristics

#### 4.3.1. Hospital Ownership

Higher mortality rates in government, military, and state-owned hospitals compared with private hospitals highlight disparities in resources, infrastructure, and care quality, with studies showing public hospitals often have higher adjusted mortality rates than private not-for-profit hospitals [35]. This suggests that ownership type may be associated with variations in mortality rates, potentially due to differences in hospital resources, staff qualifications, or patient care practices. A previous study shows that both teaching and non-teaching public hospitals have higher mortality rates than private teaching hospitals [36]. This could be due to the quality of care, which is suggested to be associated with hospital type, and the fact that private teaching hospitals may have more specialized staff and advanced technologies. A systematic review also found that private for-profit hospitals have a higher risk of death than private not-for-profit hospitals [37]. This could imply that the profit motive in for-profit hospitals might lead to cost-cutting measures that could adversely affect patient care, although this does not directly compare government-owned with privately owned hospitals. Research in veterans affairs hospitals shows they have higher mortality rates than university medical centers, suggesting systemic issues in government facilities [38]. Another study suggests that unadjusted mortality rates were lower in for-profit facilities compared with government-owned facilities [39]. However, when adjusted for variables such as the rate of discharge to general hospitals, these differences disappeared, indicating that operational practices such as discharge policies might influence mortality rates.

In summary, previous studies indicate that higher in-hospital mortality rates in government-owned hospitals compared with private-owned hospitals may be influenced by factors such as the quality of care, resources, staff qualifications, operational practices, and potentially the presence of a profit motive. Meanwhile, in Indonesia, during the era of the National Health Insurance (JKN) program, it is common to encounter patients with mild symptoms seeking care at private hospitals and making out-of-pocket payments or utilizing private insurance. Conversely, in severe cases, private hospitals tend to refer patients to government-owned hospitals. This phenomenon may contribute to higher in-hospital mortality rates in government-owned hospitals compared with private hospitals.

#### 4.3.2. Hospital Location

In this study, geographical factors such as city or county settings and regional variations emerge as significant determinants of disparities in diabetes mellitus among inpatients, elucidating distinctions in disease prevalence, management strategies, and outcomes across diverse regions and among various racial and ethnic demographics. The prevalence of diabetes among inpatients with acute coronary syndrome (ACS) varies geographically within China, with higher rates observed in the northeast and lower rates in the southwest. Similarly, awareness and treatment rates of diabetes differ significantly across regions [40]. Another study indicates a decrease in diabetes-related hospitalizations from 2008 to 2016/2017, and state-level data reveal disparities, particularly among young adults and in rural areas. However, no significant differences were found among racial/ethnic groups [41].

The disparities in mortality rates among different hospital regions in Indonesia underscore potential inequalities in healthcare access, infrastructure, and delivery models. These findings reflect broader discussions on regional healthcare disparities shaped by socioeconomic, cultural, and geographical factors. Addressing these issues requires effective local governance and health decentralization to ensure equitable healthcare quality nationwide. There is a notable lack of comprehensive literature on healthcare quality in Indonesia, particularly concerning diabetes mellitus in-hospital mortality. Existing studies primarily focus on visit rates or healthcare utility, neglecting critical aspects like quality of care and patient outcomes [42]. This knowledge gap impedes understanding of the country’s healthcare system’s effectiveness in managing severe chronic conditions.

Indonesia being an archipelago leads to significant geographic disparities affecting healthcare quality and accessibility. Variations in life expectancy and maternal and infant mortality rates reveal an unequal distribution of health resources. For instance, Java Island, being more developed, has more hospitals and specialist doctors compared with remote areas [43]. Consequently, there are disparities in healthcare access, with some regions facing shortages of essential services. Moreover, disparities manifest in the concentration of hospitals and specialists, mainly on Java Island, while referral hospitals are typically located in provincial capitals and are inaccessible to remote residents. This necessitates long travel distances for patients seeking care, often exacerbating their conditions owing to treatment delays [44].

### 4.4. Study Limitations

A limitation of this research paper is the potential for selection bias in the sample population. The study utilizes claim data from the Social Security Health Insurance Program (JKN) in Indonesia, which are openly accessible. However, the sample selection process may introduce bias, as it relies on participants who have joined the program before 2022 and those who joined in 2022. This method may not capture the entire population, particularly those who did not participate in the program or those who joined after 2022. Therefore, the findings may not be fully representative of the entire population of individuals with diabetes mellitus in Indonesia.

Additionally, the reliance on claim data may introduce limitations in the accuracy and completeness of the information. Claim data may be subject to errors in coding, documentation, or reporting, which could affect the validity of the results. Furthermore, the study focuses on in-hospital mortality among patients with type 2 diabetes mellitus, which may not fully capture mortality occurring outside of the hospital setting or among individuals with other types of diabetes.

Moreover, while the study examines various patient and hospital characteristics associated with in-hospital mortality, it does not account for potential confounding variables that may influence the outcomes. Factors such as socioeconomic status, access to healthcare services, lifestyle factors, and adherence to treatment regimens could confound the associations observed in the study.

Lastly, the study’s observational design limits the ability to establish causal relationships between the independent variables and in-hospital mortality. While multivariable regression analysis is conducted to examine the associations, it cannot infer causality, and there may be unmeasured confounders or residual confounding that could affect the results. Therefore, caution should be exercised when interpreting the findings as causal relationships.

### 4.5. Future Research Directions

Conduct longitudinal studies to explore the trajectories of patients with non-dependent insulin diabetes mellitus beyond the hospitalization period, assessing long-term outcomes, healthcare utilization patterns, and factors influencing post-discharge morbidity and mortality.

Investigate the quality of care delivery across different hospital types and regions to identify modifiable factors contributing to disparities in health outcomes. Assess interventions aimed at improving healthcare access, equity, and quality to reduce mortality differentials.

Utilize precision medicine approaches to identify patient-specific risk factors, biomarkers, and therapeutic targets for optimizing treatment outcomes and reducing mortality rates in patients with non-dependent insulin diabetes mellitus.

Evaluate the impact of healthcare policies, financing mechanisms, and regulatory frameworks on mortality outcomes among diabetic patients. Investigate the effectiveness of interventions targeting social determinants of health and health equity in reducing disparities in healthcare outcomes.

## 5. Conclusions and Recommendations

### 5.1. Conclusions

In conclusion, this study provides valuable insights into the factors influencing in-hospital mortality among patients with non-dependent insulin diabetes mellitus. Addressing identified risk factors, improving access to quality healthcare, and implementing region-specific interventions are crucial steps toward reducing mortality rates and improving outcomes in this vulnerable population.

### 5.2. Recommendations

Based on the findings presented, we suggest the following policy recommendations for reducing mortality rates:Addressing gender disparities: Policies should focus on addressing gender-specific healthcare needs and improving access to healthcare services for female patients. Efforts should be made to understand and mitigate factors contributing to higher mortality rates among females.Targeted interventions for older patients: Healthcare policies should prioritize interventions aimed at managing age-related health deterioration, including the management of comorbidities and the provision of specialized care for older patients.Support for informal workers and subsidized groups: Policies should aim to improve healthcare access and quality for informal workers and subsidized groups, recognizing their higher risk of mortality and addressing barriers to healthcare utilization.Enhancing hospital levels and regions: Efforts should be made to strengthen healthcare infrastructure and services in regions with higher mortality rates, such as Sulawesi, Kalimantan, Sumatera, Papua, Nusa Tenggara, and Maluku. Additionally, attention should be given to improving the quality of care in higher-level hospitals (Classes A, B, and C) and specialist services hospitals to reduce mortality rates.

Healthcare policies should prioritize interventions aimed at addressing patients with severe cases, as they demonstrate markedly heightened odds of mortality. Such interventions may encompass specialized care, early identification of high-risk individuals, and intensive monitoring. Additionally, recommendations for primary prevention, including screening and early diagnosis, should be considered beforehand. Thus, cases with high severity levels may decrease along with in-hospital mortality.

Efforts should be made to strengthen healthcare delivery systems, improve access to healthcare services, and enhance the quality of care across all levels of hospitals and regions. This may involve investments in healthcare infrastructure, workforce training, and the implementation of evidence-based clinical practices.

By implementing these policy recommendations, stakeholders can work toward reducing mortality differentials among hospitalized patients, particularly those with non-dependent insulin diabetes mellitus, and improving overall healthcare outcomes.

## Figures and Tables

**Figure 1 ijerph-21-00581-f001:**
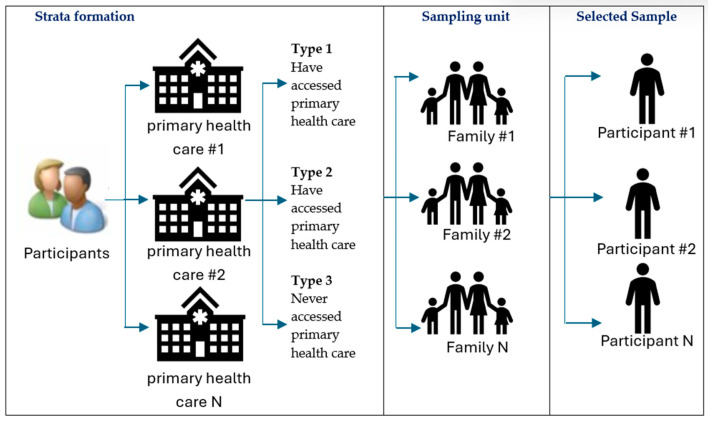
Sampling Step Illustration.

**Figure 2 ijerph-21-00581-f002:**
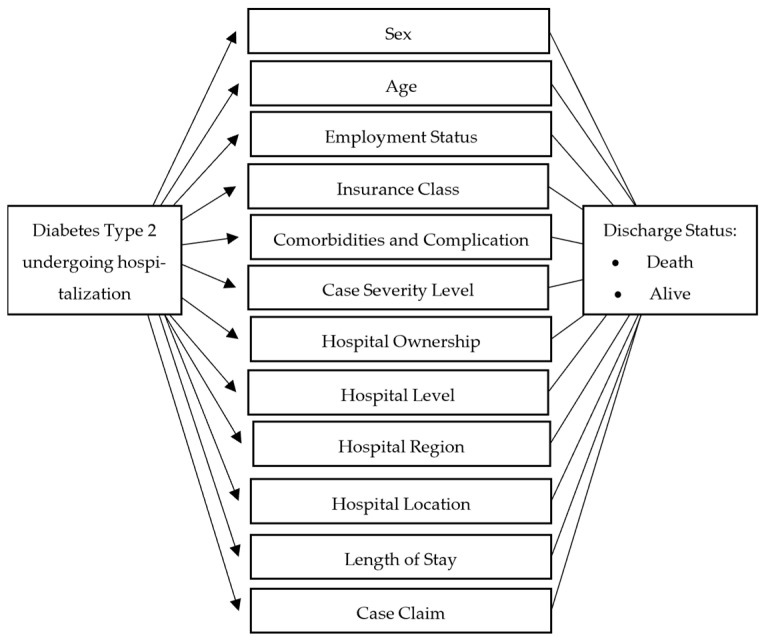
Conceptual Framework.

**Table 1 ijerph-21-00581-t001:** Illustration of Weight Calculation for Each Household in Six Strata.

No	Primary Healthcare ID	Family Categories	Population Size of Family	Number of Family Samples	*p_i_* (Likelihood of Being Selected)	*w_i_* (Weight)
1	1	1	150	1	0.067	150
2	1	2	400	1	0.025	400
3	1	3	6200	1	0.002	6200
4	2	1	200	1	0.050	200
5	2	2	500	1	0.020	500
6	2	3	5900	1	0.002	5900

**Table 2 ijerph-21-00581-t002:** Primary outcomes, samples, and hospital characteristics.

Variable	Categories	*n*	%
Outcome Variable			
Hospital discharge status	Alive	570.458	93.4
	Dead	40.351	6.6
Patient Characteristic			
Sex	Male	233.585	38.2
	Female	377.224	61.8
Age	Maximum	96	
	Minimum	30	
	Median	57	
Employment Status	Dependent Employment	138.488	22.7
	Independent Employment	245.916	40.3
	Subsidized group	226.405	37.1
Insurance Class	Class 1	170.960	28.0
	Class 2	117.192	19.2
	Class 3	322.657	52.8
Comorbidities			
Disorders of fluid, electrolyte and	No	547.921	89.7
acid–base balance (ICD10:E87)	Yes	62.888	10.3
Anemia (ICD10:D64)	No	584.956	95.8
	Yes	25.853	4.2
Urinary tract infection (ICD10:N39)	No	579.117	94.8
	Yes	31.692	5.2
Pneumonia (ICD10:J18)	No	588.484	96.3
	Yes	22.325	3.7
Complication (diagnose on discharge)			
NIDDM without complications	(ICD10:E119)	610.809	75.6
NIDDM with coma	(ICD10:E110)	461.522	4.6
NIDDM with ketoacidosis	(ICD10:E111)	28.316	3.7
NIDDM with multiple complications	(ICD10:E117)	22.340	4.9
NIDDM with other specified complications	(ICD10:E116)	30.001	7.3
NIDDM with unspecified complications	(ICD10:E118)	44.696	3.9
Case Severity Level	Mild	365.326	59.8
	Moderate	158.147	25.9
	Severe	87.336	14.3
Hospital Characteristic			
Hospital Ownership	Government	247.086	40.5
	Military/ Police	36.557	6.0
	State-Owned Enterprise (BUMN)	20.190	3.3
	Private	306.976	50.3
Hospital Level	A	20.822	3.4
	B	183.581	30.1
	C	286.724	46.9
	D	109.223	17.9
	Specialist services	10.459	1.7
Hospital Region	Jawa and Bali	374.397	61.3
	Sumatera	130.231	21.3
	Sulawesi	45.196	7.4
	Kalimantan	34.496	5.6
	Papua, Nusa Tenggara, Maluku	26.489	4.3
Hospital Location	City	275.020	45.0
	County	335.789	55.0
Length of Stay	Maximum	46	
	Minimum	0	
	Median	4	
Case Claim (in USD)	Maximum	1089	
	Minimum	147	
	Median	300.09	

**Table 3 ijerph-21-00581-t003:** Comparison of patient and hospital characteristics with in-hospital mortality rates.

Variables	Categories	Alive		Dead	
		*n*	%	*n*	%
Patient Characteristic					
Sex	Male	218.898	93.7	14.687	6.3
	Female	351.560	93.2	25.664	6.8
Employment Status	Dependent Employment	133.291	96.2	5.197	3.8
	Independent Employment	231.128	94.0	14.788	6.0
	Subsidized group	206.039	91.0	20.366	9.0
Insurance Class	Class 1	162.408	95.0	8.552	5.0
	Class 2	111.377	95.0	5.815	5.0
	Class 3	296.673	91.9	25.984	8.1
Comorbidities					
Disorders of fluid, electrolyte and	No	514.034	93.8	33.887	6.2
acid–base balance (ICD10:E87)	Yes	56.424	89.7	6.464	10.3
Anemia (ICD10:D64)	No	546.275	93.4	38.681	6.6
	Yes	24.183	93.5	1.670	6.5
Urinary tract infection	No	540.756	93.4	38.361	6.6
(ICD10:N39)	Yes	29.702	93.7	1.990	6.3
Pneumonia (ICD10:J18)	No	550.000	93.5	38.484	6.5
	Yes	20.458	91.6	1.867	8.4
Complication					
NIDDM without complications	(ICD10:E119)	441.214	95.6	20.308	4.4
NIDDM with coma	(ICD10:E110)	21.549	76.1	6.767	23.9
NIDDM with ketoacidosis	(ICD10:E111)	15.693	70.2	6.647	29.8
NIDDM with multiple complications	(ICD10:E117)	25.938	86.5	4.063	13.5
NIDDM with other specified complications	(ICD10:E116)	43.553	97.4	1.143	2.6
NIDDM with unspecified complications	(ICD10:E118)	22.511	94.1	1.423	5.9
Case Severity Level	Mild	351.991	96.3	13.335	3.7
	Moderate	149.082	94.3	9.065	5.7
	Severe	69.385	79.4	17.951	20.6
Hospital Characteristic					
Hospital Ownership	Government	223.843	90.6	23.243	9.4
	Military/Police	34.271	93.7	2.286	6.3
	State-Owned Enterprise (BUMN)	20.025	99.2	165	0.8
	Private	292.319	95.2	14.657	4.8
Hospital Level	A	18.962	91.1	1.860	8.9
	B	167.563	91.3	16.018	8.7
	C	269.721	94.1	17.003	5.9
	D	105.308	96.4	3.915	3.6
	Specialist services	8.904	85.1	1.555	14.9
Hospital Region	Jawa and Bali	349.145	93.3	25.252	6.7
	Sumatera	123.454	94.8	6.777	5.2
	Sulawesi	41.101	90.9	4.095	9.1
	Kalimantan	31.043	90.0	3.453	10.0
	Papua, Nusa Tenggara, Maluku	25.715	97.1	774	2.9
Hospital Location	City	260.406	94.7	14.614	5.3
	County	310.052	92.3	25.737	7.7

**Table 4 ijerph-21-00581-t004:** Determinants of non-dependent insulin diabetes mellitus in-hospital mortality.

Variable	Categories	*p* Value	Odd Ratio (CI 95%)
Patient Characteristic			
Sex	Male	reference	
	Female	>0.05	
Age		<0.05	1.03 (1.03–1.03)
Employment Status	Dependent Employment	reference	
	Independent Employment	<0.05	0.89 (0.86–0.93)
	Subsidized group	<0.05	1.84 (1.75–1.94)
Insurance Class	Class 1	reference	
	Class 2	>0.05	
	Class 3	>0.05	
Complications	NIDDM without complications (ICD10:E119)	reference	
	NIDDM with coma (ICD10:E110)	<0.05	4.48 (4.32–4.64)
	NIDDM with ketoacidosis (ICD10:E111)	<0.05	10.86 (10.46–11.28)
	NIDDM with multiple complications (ICD10:E117)	<0.05	3.56 (3.42–3.70)
	NIDDM with other specified complications (ICD10:E116)	<0.05	0.74 (0.69–0.78)
	NIDDM with unspecified complications (ICD10:E118)	<0.05	1.74 (1.64–1.84)
Comorbidities	Disorders of fluid, electrolyte, and acid–base balance (ICD10:E87)	<0.05	1.04 (1–1.07)
	Anemia (ICD10:D64)	<0.05	0.53 (0.5–0.56)
	Urinary tract infection (ICD10:N39)	<0.05	0.51 (0.48–0.53)
	Pneumonia (ICD10:J18)	<0.05	0.30 (0.28–0.32)
Case Severity Level	Mild	reference	
	Moderate	<0.05	1.93 (1.84–2.01)
	Severe	<0.05	10.49 (9.78–11.24)
Hospital Characteristic			
Hospital Ownership	Private	reference	
	Government	<0.05	1.27 (1.23–1.3)
	Military/Police	<0.05	1.48 (1.4–1.57)
	State–owned enterprises (BUMN)	<0.05	0.19 (0.16–0.22)
Hospital Level	D	reference	
	A	<0.05	2.60 (2.33–2.88)
	B	<0.05	2.31 (2.21–2.4)
	C	<0.05	1.53 (1.46–1.58)
	Specialist Services	<0.05	4.39 (4.09–4.72)
Hospital Region	Jawa and Bali	reference	
	Sumatera	<0.05	1.27 (1.23–1.31)
	Sulawesi	<0.05	2.36 (2.27–2.46)
	Kalimantan	<0.05	1.75 (1.67–1.83)
	Papua, NusaTenggara, Maluku	<0.05	0.42 (0.39–0.46)
Hospital Location	City	reference	
	County	<0.05	1.49 (1.44–1.52)
Length of Stay		<0.05	0.81 (0.8–0.81)
Case Claim (in USD)		>0.05	

## Data Availability

Available from https://data.bpjs-kesehatan.go.id of the Health Social Security Agency (BPJS Kesehatan), accessed on 27 December 2023.

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
