# Peer review of "Behind the Hospital Ward: In-Hospital Mortality of Type 2 Diabetes Mellitus Patients in Indonesia (Analysis of National Health Insurance Claim Sample Data)"

_ijerph, 2024, doi:10.3390/ijerph21050581_

Round 1
Reviewer 1 Report
Comments and Suggestions for Authors
This study investigates in-hospital mortality rates among 610,809 hospitalized patients with type 2 diabetes mellitus (T2DM) using data from Indonesia's National Health Insurance Program. These findings provide valuable insights for healthcare practitioners, policymakers, and researchers in understanding and addressing the complex interplay of factors influencing T2DM outcomes. However, few points need to be considered.
The analysis reveals a mortality rate of 6.6%, with factors such as female gender associated with increased mortality risk. However, the report lacks an explanation for the observed gender disparity.
it's noted that while the study identifies independent key factors influencing T2DM outcomes, important variables such as socioeconomic status, lifestyle factors, medication adherence, genetic predisposition, and psychosocial factors were not included in the analysis. Considering these variables could provide a more comprehensive understanding of diabetes outcomes and inform targeted interventions for improved patient care.
Comments on the Quality of English LanguageGood
Reviewer 2 Report
Comments and Suggestions for Authors
The study is original and clearly formulated; the important task is to study the factors that determine in-hospital mortality in patients with type 2 diabetes.
The introduction is well written, allowing a comprehensive look at the features of economic development and healthcare in Indonesia, the stages of medical care, which, in the aspect of the task set by the researchers, is important introductory information that allows us to understand further conclusions.
The data is reliable enough to draw conclusions. The article is clear and interesting, it should attract a wide readership.
The strength of the study is the large sample of patients (610,809 patients) and correct statistical processing.
The article is well structured, allowing for a sequential discussion of each of the factors potentially influencing in-hospital mortality.
There are a number of comments:
1. The title of the article reads Non Insulin Dependent Diabetes Mellitus, in the abstract and further in the discussion it sounds like type 2 diabetes, which is more correct from the point of view of modern requirements for diagnosis. (although Non Insulin Dependent Diabetes Mellitus is found in the international classification of diseases).
2. In the description of materials and methods there is no clear information at all that they specifically took patients with type 2 diabetes. At the same time, there are no other complaints about this section - methods, tools, and statistical processing are described in sufficient detail.
3. Unfortunately, among the analyzed factors there is no indication of how common complications of diabetes mellitus were (coding according to ICD10 assumes the indication of Non Insulin Dependent Diabetes Mellitus with eye damage, ... with kidney damage, ... with multiple complications, ... with unspecified complications etc. It is a pity that this information is not provided.
4. “The increasing global prevalence of diabetes mellitus, a chronic metabolic disorder, poses significant challenges to healthcare systems around the world.” - indicate that this is type 2 diabetes, because This type has become widespread worldwide.
5. In Table 3. Determinants of Non-Dependent Insulin Diabetes Mellitus In-Hospital Mortality, several features (Dyspepsia Hyper lipoprotein/ lipidaemias - see lines below) have an odds ratio of less than one, which means that their presence is associated with a reduced risk of mortality. However, in the description of the results there is only a mention that these factors are associated with altered mortality rates. I would like a more detailed discussion of these factors.
Table 3
Dyspepsia (ICD10: K30) 0.05 0.2 (0.19-0.22)
Hyper lipoprotein/ lipidaemias (ICD10: E78, excl E78.6) 0.05 0.27 (0.24-0.3)
Further in the discussion section 4.3. I also did not see any discussion of comorbidity factors. Although it should be noted that concomitant diseases in general are considerations when discussing the characteristics of women.
At the same time, what is stated in the article “providing practical information to practicing physicians to optimize patient care protocols and improve clinical outcomes” does not allow solving this problem with this study design, because treatment protocols are not evaluated, but only factors and their association with mortality among patients with diabetes are taken into account.
The literature sources are generally described and selected satisfactorily. 52.2% of sources are not older than 10 years, 26.1% of sources are not older than 5 years, 43.5% of sources are older than 10 years.
Comments on the Quality of English LanguageIt is recommended to show the manuscript to a native speaker
Reviewer 3 Report
Comments and Suggestions for Authors
1. Main question addressed by the research
T2DM can lead to a range of severe complications, including cardiovascular diseases, diabetic retinopathy, diabetic neuropathy, and kidney complications. Increased mortality rates related to diabetes have also been observed in young individuals. The study aims to examine the incidence of in-hospital mortality among hospitalized patients with type 2 diabetes mellitus and to identify the determinants of this outcome. The goal is to provide actionable information to optimize care protocols and improve clinical outcomes for patients with this condition.
The focus of the study is Indonesia. Indonesia is a middle-income country with a significant mortality rate associated with diabetes, which has become one of the leading causes of death. The number of diabetes cases is rapidly growing, with a high number of undiagnosed cases. Indonesia has implemented a national health insurance program to ensure access to healthcare services for all citizens. The program covers medical expenses, including specialized services for patients with T2DM.
2. Field Relevance
The study utilizes claim data from the Social Security Administration (BPJS) Kesehatan, providing insights into T2DM within the Indonesian population. This approach allows for a comprehensive analysis of patient characteristics, hospital outcomes, and mortality determinants. Moreover, the paper focuses on in-hospital mortality among patients with T2DM, which is a crucial aspect of diabetes management often overlooked in research studies. Understanding the factors contributing to in-hospital mortality can help improve patient care protocols and clinical outcomes. The study employs multivariable regression analysis to explore the determinants of in-hospital mortality. This statistical approach allows for a comprehensive examination of the relationship between various patient and hospital characteristics and mortality outcomes. The paper identifies several determinants associated with in-hospital mortality, including patient characteristics such as gender, age, employment status, and comorbidities, as well as hospital characteristics such as ownership, level, and location. By elucidating the factors associated with in-hospital mortality, the study aims to provide actionable information for healthcare practitioners to optimize patient care protocols and improve clinical outcomes. This can lead to enhanced quality of care delivery and ultimately improve patient survival rates.
3. Scientific addition to the field
The paper addresses a significant gap in the field by focusing on in-hospital mortality among patients with type 2 diabetes mellitus. While much research has been conducted on the epidemiology, management, and long-term complications of diabetes mellitus, there is relatively limited literature specifically examining in-hospital mortality and its determinants. Understanding the factors contributing to in-hospital mortality is crucial for developing targeted interventions to improve patient outcomes and reduce mortality rates among individuals with diabetes mellitus.
4. Improvements
While the methodology of the study appears robust, there are several specific improvements and further controls that the authors could consider:
- The study utilizes claim data from 2017 to 2022, which allows for a longitudinal analysis of trends over time. However, the authors could provide more detailed information about how they accounted for potential changes in data collection methods, healthcare policies, or patient demographics over the study period.
- The authors mention using a stratified random sampling method for sample selection. It would be beneficial to provide more information on how the samples were stratified and whether there were any biases introduced during the selection process.
- The authors mention using a weighting variable in their analysis to ensure that the obtained results represent the population. They should provide more details on how this weighting variable was calculated and whether it effectively corrected for any biases or differences between the sample and the population.
- The authors should provide more information on the specific regression models used, including the inclusion of interaction terms or nonlinear relationships if relevant. Additionally, they could consider conducting sensitivity analyses to assess the robustness of their findings to different model specifications.
- In addition to the variables examined in the analysis, the authors should consider including additional control variables that may confound the relationship between patient/hospital characteristics and in-hospital mortality.
- Given the diverse nature of the patient and hospital characteristics examined, the authors could consider conducting subgroup analyses to assess whether the associations between these factors and in-hospital mortality vary across different patient populations or healthcare settings. This would provide more nuanced insights into the determinants of mortality in patients with T2DM.
5. Conclusions of study
The conclusions drawn in the study are generally consistent with the evidence and arguments presented throughout the paper and address several key factors associated with in-hospital mortality among patients with non-dependent insulin diabetes mellitus, including sex, age, employment status, hospital characteristics, comorbidities, and case severity. Overall, the conclusions drawn in the study are supported by the evidence presented, addressing the main questions posed regarding factors influencing in-hospital mortality among patients with non-dependent insulin diabetes mellitus. The conclusions provide valuable insights into the determinants of mortality and highlight areas for further investigation and intervention in diabetes care.
6. Reference appropriation: Fine
7. Comments on tables, figures, data
The tables and figures presented in the study provide valuable information on patient characteristics, hospital characteristics, and factors associated with in-hospital mortality among patients with non-dependent insulin diabetes mellitus. Overall, the tables and figures enhance the clarity and interpretability of the study findings, while the quality of the data appears adequate for addressing the research objectives.
Comments on the Quality of English LanguageMinor editing necessary, please check for typo errors.
Round 2
Reviewer 1 Report
Comments and Suggestions for Authors
The authors have made substantial changes in several part of the paper and addressed the reviewers’ comments. This manuscript may be accepted for publication.